# Considerations for Children's Nature Connection and Park Environmental Justice in Western Societies

**Melissa VanSickle * and Christopher Coutts** 

Department of Urban and Regional Planning, Florida State University, 330 Bellamy Building, Tallahassee, FL 32306, USA
* Correspondence: mv09f@fsu.edu

**Abstract:** Nature provides multiple physical, mental, and social health benefits to children. Although green spaces in cities can provide these health benefits, in many Western societies, children are spending less time outdoors and access to green spaces and related factors are not evenly distributed spatially and demographically. In addition, recent literature points to new greening projects furthering inequities due to processes of green gentrification. Several studies have provided insight into distributional, interactional, and procedural environmental justice issues related to green spaces. Through a narrative review of the literature, this paper explores these components of environmental justice as they relate to children's nature connection and play in local city parks. When planning for local parks, historical and context-specific social and environmental factors that influence caretaker and children's ability to access green spaces that promote nature connection should be considered.

**Keywords:** parks; environmental justice; children; nature connection

## 1. Introduction

Our environments influence human health in many ways; thus, across the globe, the intersection between the practices of urban planning and public health have been essential in creating more equitable and sustainable places [1]. Outlining the relationships between the natural environment and human health, an ecosystem service framework, as described by the Millennium Ecosystem Assessment (MA) [2] provides four pathways to human wellbeing. These pathways include provisioning, regulating, cultural, and supporting services [2].To address these aspects of health, locally planned green infrastructure projects and initiatives, such as the development of local city parks, seek to increase access and connection to nature [3]. For example, cities across the U.S. and Europe have implemented various types of brownfield redevelopment and green infrastructure projects, which aim to promote more access green space [4].

Yet, as discussed in recent green cities and social injustice literature, environmental justice research must consider not only inequities in the exposure to hazardous environmental conditions, but also in the distribution of these health producing environments, such as green spaces [5]. More recently, due to a sustainable green turn in urban planning and design, there are additional concerns surrounding the implementation of new greening projects and associated processes of green gentrification, which results in the furthering of social and environmental inequities [6,7]. Thus, across the globe, initiatives related to urban green justice aim to not only increase the amount of green spaces, but also consider the multiple interrelated inequities that exist within these development processes. For example, in the U.S. and Europe, there are several examples of both governmental and community-led strategies which aim to address the complex issues related to green space policy, design, and their embeddedness within additional economic and socio-cultural landscapes [8].

Urban green spaces, such as local city parks, can provide positive well-being benefits, promoting healthy development from childhood into adulthood [9]. Unlike other ecosystem

services, cultural ecosystem services (CES) are place-specific, and their production relies on both social and environmental factors [10]. One mechanism contributing to this nature–health relationship is the CES of nature connection, a subjective sense of one's relationship with nature which varies across context and time [11]. Childhood nature connection has been found to be associated with adult nature connectedness, pointing to concepts related to a critical period for developing biophilia [12]. In addition to nature connection in childhood, outdoor green spaces serve as places that promote children's engagement in adventurous or creative play [13]. These services are important since children's free play is associated with learning and development, identity formation, and sense of belonging [14] (p. 295). Correspondingly, through assessing the needs for child-friendly spaces in South Africa, Cilliers and Cornelius [15] suggest that cities should consider interactions with nature and free play along with intersecting social, physical, and economic objectives which further promote play among children.

Although nature connection and outdoor play can provide health and development benefits, there is a concern in Western societies that children today are spending less time playing outdoors than previous generations [16]. For example, city development initiatives and adult perceptions of "adequate spaces" for children, can affect children's ability to explore urban environments and engage in free play [17]. Further, access to these spaces can vary, depending on differing social and environmental contexts [18,19]. Access to and inclusion in green spaces have been found to differ across socioeconomic, racial, ethnic, age, gender, and ability lines [19]. A survey conducted by the United Nations Children's Fund (UNICEF) of 35,000 children from 65 different countries found that one in four noted a feeling of being unsafe in their city's parks, and half indicated that there are children in their city who suffer from discrimination. In addition, less than 30% felt as though their voices were being heard by the authorities [20]. Differences in social contexts or factors and their influence on nature experiences have also been highlighted in studies considering children's nature connection and play during the COVID-19 pandemic [21–23]. During COVID-19 there may have been various influences on children's play and connection to nature, yet the impacts are not experienced the same by all children. For example, one study found that affluent children in England were more likely to see increases in nature connection during the pandemic than less affluent children, pointing to a need to further understand the interrelated factors that influence children's activities and nature connection [23].

Using a socio-ecological model of environmental justice, Rigolon et al. [24] outlines interrelated distributive, procedural, and interactional justice considerations related to recreational spaces. Expanding on this framework, this paper explores these components of environmental justice as they relate to children and locally planned public parks. This framework provides a method for considering both physical and social aspects of children's park access. To explore the literature, this paper discusses these themes of environmental justice in relation to the assessment of broader social and political contexts influencing children's access and play in nature. The paper concludes with a discussion on children's participation and methods for capturing children's knowledge as it relates to the planning of public green spaces for children's play.

*A Framework for Park Environmental Justice*

Outlining different aspects of environmental justice, specifically related to parks and recreation, Rigolon, et al. [24] applies a socio-ecological model, situating common environmental justice categories such as distributive, procedural, and interactional justice in the context of various interrelating factors of policy, and the physical, perceived, social, and individual environment [24]. This model suggests interrelationships between these environments and the importance of applying a broader lens when considering aspects of recreational justice [24]. Following this model, Enssle and Kabisch [25] explored factors related to green space visitation patterns among older adults in Berlin, Germany. The study highlighted the significance of social networks, suggesting that "urban planning needs to consider both the physical and the social environment" [25] (p. 42). Distributive justice

includes ensuring equitable distribution in green spaces' availability and accessibility [25]. Interactional justice, in addition to recognition justice [26], expands upon the distribution of green spaces and includes considering multiple needs and preferences, recognizing difference, and creating a non-discriminatory environment [25,27]. Influencing both distributive and intersectional justice, procedural justice involves ensuring that people are included in the planning process through active participation [25]. In addition to these findings, there is a need to further understand the factors related to children's green space access and use. Each category of environmental justice as discussed above is considered within the context of children's local access to public outdoor parks and play spaces.

## 2. Materials and Methods

Since the fall of 2020, this paper has evolved and combined multiple literature reviews completed by the authors over a two-year period. Thus, this paper applies a narrative review approach, including the identification of literature within various topics of green space or park planning, environmental justice, children's play and nature connection, and city planning processes with or for children. Following narrative reviews such as Dachaga and de Vries [28], this approach was applied to allow for a broader search across disciplines and to combine perspectives related to the topic of children's nature connection and park environmental justice [29].

In addition, as outlined in a review by Schröder et al. [30], the first step of the review included identifying the paper's conceptual approach. The initial search of the literature performed in 2020 formed the conceptual approach to the paper through identifying common environmental justice frameworks. Google Scholar and FSU OneSearch databases were used to identify peer-reviewed literature articles. These databases were chosen due to their accessibility to the authors and ability to search journals across disciplines. Main keywords, including "green space" and "environmental justice" were used, and only articles written in English and readily accessible through the chosen databases were included in the review. Through this process, a specific framework was identified by narrowing down the literature to an article which proposed a framework of environmental justice as it specifically relates to socio-ecological factors, with a focus on parks and cultural ecosystem services. As described above, this framework provided the thematic categories that serve as the foundation of this review.

Following the conceptual framework of park environmental justice as put forth by Riglon et al. [24], the databases were searched and reviewed in the course of 2020–2022 through an iterative process which allowed for the authors to develop connections and themes across the literature. Additional keywords such as "children," "cultural ecosystem services," "parks," "nature connection," and "play" were used to identify additional specific studies and review articles. Articles were reviewed and selected based on (1) their discussion of children's access to parks, (2) their discussion of social factors which relate to children's access and play in parks, and (3) their discussion of children's participation in outdoor play space planning processes, and how these studies and reviews relate to the three main categories of environmental justice: distributive, intersectional, and procedural justice. The researcher created reference lists from selected articles that were used to identify additional peer-reviewed literature corresponding to each thematic category (using an opportunistic snowball method). A summation of key findings is discussed below to further explore the themes within the literature.

Although the review approach chosen for this paper provided the ability to connect topics from a broad range of literature, there are many limitations to this approach. The first and foremost limitation is that due to a lack of systematic methods, this review cannot be replicated. In addition, the lack of pre-specified selection criteria leaves room for bias in the selection and review of articles. In addition, this paper included searches within two databases accessible to the authors and was limited to English-language articles, which significantly limits the range of content included in this review.

## 3. Results

### 3.1. Distributive Justice

Inequity in access to quality green spaces can often be related to additional social inequities, such as those in the U.S. which follow historic and persisting processes of racial segregation and oppressions, and therefore the spatial divisions within cities along racial, ethnic, and socioeconomic lines [31,32]. As described by environmental justice literature, the underlying drivers of environmental inequality are related to interrelated systems of the market economy and institutionalized racism [33]. Further, through the commodification of nature, green space is often unevenly distributed along social and spatial lines separating those who can afford green private property versus those who rely on public green spaces [34]. Natural features such as street trees and higher quality parks correlate with higher housing prices and property values, therefore resulting in households able to pay to access quality green spaces [34–36]. These processes are also influenced by development patterns in cities. For example, green space accessibility in Shanghai, China has been associated with housing prices but also with the city's spatial development, seen through gaps in green space accessibility between the central urban areas and peri-urban areas [37].

In relation to outdoor play spaces, a study of a large sample of children ages 0–17 in the United States found that almost a quarter lacked access to a neighborhood park, while children living in poverty regardless of race or urban setting were more likely to lack access, and Black non-Hispanic children living in urban areas were more likely to lack access regardless of income [38]. Rigolon and Flohr [39], when comparing low-income and high-income neighborhoods in Denver, Colorado, found that lower-income residents had less access to higher quality parks. In the United Kingdom and New Zealand, barriers to low-income, multi-ethnic children's activity in green spaces were also associated with uneven availability of amenities such as equipment, toilets, and safe areas for children's play [18,40].

Recent literature has also described a green paradox, related to the implementation of local greening projects that claim to address inequities, yet initiate processes of green gentrification, with resulting negative effects more often experienced by low-income and Black, Indigenous, and People of Color (BIPOC) residents [19,41,42]. Further, neighborhood changes associated with gentrification may include displacement and social exclusion through newer green space projects increasing housing costs and general affordability in neighborhoods [31]. Rigolon and Németh [43], looking at multiple cities in the United States, noted that certain park features such as its type and location, including locations closer to downtown areas or projects associated with active transportation, served as triggers for gentrification. Different stages or processes of gentrification may cause varying effects that influence children's ability to experience the benefits of these new green spaces. In neighborhoods experiencing green gentrification in Barcelona, Spain, Oscilowicz [44] noted that in the short term, children may experience increased access and use of green spaces, but in the long term, neighborhood change may result in decreased feelings of trust, satisfaction, and sense of security.

Many studies related to children's access to green space consider equity in terms of its availability, quality, or proximity, yet green space access and use determinants are intertwined with various structural and community inequities, and require an expanded approach to understand justice issues as they relate to its distribution. Local governments often overlook social inclusion and largely focus on aspects of physical access [45]. For example, in a scoping review related to children with disabilities' access to outdoor play spaces, it was found that studies focused on physical access but failed to reveal or recognize the aspects of the social environment [46].

To further understand distributional inequities, it is also necessary to consider underlying historical social and political processes of exclusion [31]. For instance, access, perceptions, motivations, and green space use may be influenced by previous and current government policy, social attitudes, processes of exclusion and discrimination, and other

related processes in the provision of green space [35,36]. Accessibility therefore should be further understood as "a multidimensional concept determined by a variety of geographic, social, and economic factors" and "the relative importance of each dimension of access is context-specific" [47] (p. 443).

### 3.2. Interactional Justice

The social aspects of public parks include understanding the interactional justice aspects of children's access to outdoor play spaces. For example, new greening projects can create "sociocultural invisibilization" through processes which silence the socio-natures of historically marginalized groups [26]. Processes of gentrification can also impact identity formation and aspects of heritage-making, which can be dominated by economic development-focused greening that promotes neighborhoods for "socially and racially privileged residents" [26] (p.1745). As they relate to children's needs and preferences for connecting with natural features through "rough ground" and creative play, these preferences can also be restricted through caretaker and planners' norms of play and design [13]. Therefore, green spaces for children should respond to a diverse set of backgrounds, needs, preferences, and identities [48].

Studies considering these preferences have found that children's motivations and activities in nature and outdoor play may vary across intersections of age, gender identity, and culture, and social norms expressed through these spaces can either encourage or restrict children's activities, promoting or inhibiting their nature connection. For instance, as it relates to children's age, Chawla [13] points to research that suggests children may become more interested in social environments after the age of twelve, altering their preferences and play styles in nature. Caretaker's perceptions, concerns, and abilities also change depending on the child's age, thus influencing their use and perceptions of green space. Caretakers with young children may have barriers in caring needs, responsibilities, and fears related to their young children being in outdoor space [18]. Feng and Astell-Burt [49] also discuss differing levels of independence across children's ages, noting that younger children rely more on adults, and their experiences are more restricted by their caretaker's situations and perceptions. Green space quality may also become more relevant to children's well-being as they age [49]. For instance, older children are not only influenced by green spaces in their immediate neighborhood contexts but also by green spaces of their own choosing [50]. Additional factors that may vary among children include their nature relatedness and orientation as well as desire to use screen-based media, which Soga et al. [51] found to be factors that influence the amount of nature experiences among children in Japan.

Studies in the United States and Europe suggest that gender and culture can also relate to children's and their caretakers' preferences and activities in green spaces. A study by Reimers et al. [52] in Germany found that in addition to differences in preferred play activities, the presence of boys in these spaces limited girls' physical activity levels. Studies have also found that children's preferences and need for different green space features and amenities varied across cultural backgrounds and ethnic groups, and play was often influenced by varying environmental and economic factors [49,53]. In Europe, it has been found that children from immigrant families may be stigmatized and experience forms of exclusion, as social constructions surrounding green space have resulted in the "othering" of some residents. Public green spaces can provide social cohesion and places for children to meet and develop friendships across multiple gender or cultural identities [54,55]. Thus, there is a need to further understand ways to promote inclusion of multiple identities within these spaces, which produces the ability for all children to experience nature connection.

Although green space can provide social cohesion and other benefits to children, it is important to understand factors that influence feelings of belonging, and affect both children's and their caretakers' perceptions, circumstances, and embodied identities in these spaces [56]. Although research suggests ways in which preferences and activities vary by age, gender, or culture, these are not fully understood and can vary depending

on historical and current environmental and social contexts. Therefore, there is a need to further explore factors related to the inclusion and/or exclusion of children's identities and preferences, which influence their feelings of connection to nature. For children, these considerations may also include understanding caretakers' identities, perceptions, circumstances, and other interrelated factors, since caretakers and children can influence each other's experiences in nature [13].

### 3.3. Procedural Justice

Procedural justice involves ensuring that people are included in the planning process through active participation [25]. To address these injustices, Rigolon and Flohr [39] suggest a "bottom-up" approach, particularly as it relates to planning for greening in low-income neighborhoods, and those inhabited by historically marginalized groups. Development demands, funding constraints, and neoliberal-oriented approaches may limit equitable green space provision, and thus local knowledge and engagement of community stakeholders should be included [57]. Strategies can include local communities engaging in the production of space, such as through the creation of community gardens [39], or other forms of collective green spaces.

Since children often lack the ability to express preferences, due to norms and ideas of play, Wood [58] proposes a theory of children's spatial participation, which sees children as being active participants within space and place. Further, there is a need to recognize "how young people negotiate, challenge, and resist social control" [59] (p. 32). For example, community-led initiatives and participatory methods allow children and youth to inform the planning process, but also to be change agents and influence power dynamics [60].

There are various aspects of children's knowledge of their own experience of space that can provide insight into planning for green spaces. As discussed by Chawla [61] (p. 221), "children and youth are the experts on what fosters or fractures their personal sense of well-being." Although children's cognitive development varies by age, Freeman et al. [62] points to the research on preschool children suggesting that spatial understanding is an early development, yet adults' perceptions of children's competencies can limit children's participation in planning processes. Children's lived experience and perspective may reveal aspects of their environments often missed by adults. For example, in a study by Freeman, et al. [62], when first beginning their community mapping project, the first child noted that the study did not provide people as an option in the mapping, an important piece of the city that was left out by the adult researchers. Although this single example is not sufficiently representative of all children, it was noted by the researchers that this realization by the child had originally been overlooked by the adult researchers.

Research has also suggested that children's perceptions differ from those of adults, as they may have more experiences that are sensory-directed rather than informed by pre-constructed ideas and thinking patterns [63]. These types of studies point to the unique perspectives of children and their ability to be skilled in observing and thinking both visually and spatially [64]. Recognizing the many aspects of children's knowledge can also include methods for better understanding the ways that children's emotions shape their preferences and behaviors. Children's emotional geographies may provide a way for planners to learn how these spaces can promote experience in nature connection and play [65].

## 4. Discussion

There is a wide range of literature on children's access and experiences in green spaces related to the three categories of environmental justice included in this review. Studies often consider multiple interrelated physical and social aspects of the environment, yet analyses of underlying social and political processes are limited. In relation to children, it is important to consider the context of wider social differences, and to examine the reciprocal influence of adults in shaping children's lives [66]. For example, children's responses or conversations with adults are influenced by what is taught as "acceptable" in certain

settings [64]. In addition, it is important to consider the discourse used when engaging children. For instance, the common discourse of planning problems and solutions may not relate to children's everyday realities, and thus fails to fully engage them, or misses important aspects of their environments [67].

Narratives inform how we know what accessibility is and therefore who may be overlooked. For example, putting forth a concept of access-knowledge, Hamraie [68] argues that in post-American with Disabilities Act (ADA), accessible design has become something taken as commonsense, yet its conceptualizations and narratives should be critically analyzed to understand the influence of social values behind the construction of space. To expand upon this concept, and to further understand the social production of inequities in children's access to green spaces, studies should include further assessment of the ways in which norms are constructed and expressed through space, by considering concepts of children and play, the influence of neoliberal ideology, and adult-dominant narratives about these spaces.

### 4.1. Concepts of Children and Play

Childhood is a "historically and socially constructed concept," and the meaning and experience of it varies across intersections of identity, as well as spatial and temporal location [69,70] (p. 66). Holloway & Valentine [66] explore the invention of childhood, pointing to its relation to theories that consider uncovering the social construction of identities. In a Western context, childhood has been conceptualized as a period of becoming and a time for socialization into adulthood [66]. For example, Gillespie [70] provides a historical account of children's use of space in the United States, pointing to the social and physical ordering of the street and its interrelation with the development of adult norms about the need for children's supervision. Through these histories, children evolved into a group segregated by age, and separate from adults. Society and institutions, including urban planning, reinforced these concepts in what Gillespie [70] (p. 75) refers to as the "social and spatial regulation of children's lives."

As discussed by Goodley and Runswick-Cole [71], further research into "the normal child" would shed light on the ways in which children are judged regarding their access and play in outdoor spaces. Further, it is suggested that researchers should deconstruct the norms perpetuated in relation to children's play, development, and disability [71] (p. 510). As discussed in this review, "rhetoric, discourse, representation, history, and ideology, is crucial to addressing the gaps in commonsense understandings of access" [68] (p. 259). In addition, the ways in which children are perceived by adults can increase management by adults of children in these spaces. For example, Von Benzon [72] points to Goffman to further explain how children's opportunities are therefore limited based on how they experience outdoor spaces, and are treated within these spaces based on stigma. These social attitudes are not only experienced in one's immediate interactions but can also be expressed through public policy [72].

### 4.2. Neoliberal Ideology and Adult-Dominant Narratives

The adultification of outdoor spaces for children can lead to a more ordered environment that is spatially and socially controlled. Literature regarding children and outdoor play points to ways in which playground equipment and play spaces are often regulated by adult safety and/or liability concerns [73,74]. Wood [58] also criticizes the adult construct of norms surrounding playgrounds and of children's play behavior. Planning for "child-designated" play spaces, particularly in the U.S. and Europe, has become more controlled, surveilled, and supervised [14] (p. 295). Similarly, Pérez del Pulgar et al. [75] (p. 3) suggests that in general, development trends have "produced unsustainable, adult-centered environments" which influence children's development and wellbeing.

Further, universal, or objective narratives about the availability of outdoor play spaces and children's health "obscure the ways in which the everyday urbanism of children's green spaces actually enables or prevents construction of play and access to play" [75] (p. 4).

A study in Sweden pointed to neoliberal utopian ideals that influence adult concepts of space, and thus shape children's geographies [76]. Although there are influential structures that constrict and marginalize children in relation to decision-making processes, it is still necessary to seek how they can contribute valuable information about nature connection and challenge norms [66].

Planning for local public outdoor play spaces, such as parks, sports fields, or other recreational areas, is often either directly or indirectly determined through a city's development initiatives, which then dictate the priorities for these spaces through city policies. For example, in Glasgow, Scotland, deficiencies in green space for children's play were tied to the influence of concerns for the "politics and cultural image of the city." These concerns resulted in the prioritization of the development of larger green spaces and sports fields, which did not necessarily serve the needs of children throughout the city [77]. Further, in China, Nan [78] found that development competition and elites in cities influenced the implementation of policies related to the planning of child-friendly spaces. Similar findings by Dooling et al. [79] in the U.S. portray how citywide agendas and the economic, political, and cultural conditions tend to influence the "park planning culture" and the commodification of these spaces.

## 5. Conclusions

Inequities in children's access to and activity in green spaces are related to historical and continued processes of social stratification and exclusion; norms produced through green space design and concepts of childhood; and a lack of inclusion in planning processes and the production of space. Considering these inequities, using a socio-ecological model assists in exploring how multiple factors interrelate with both caretakers' and children's exposure to, perceptions of, and activities in green space; however, concepts of justice should be further expanded to include a critical justice framework.

To break from these narrow concepts and the ordering of and for children, Gillespie [70] suggests several avenues for incorporating children into planning theory. These avenues include examining dominant norms and how children's development is influenced by issues being addressed by planning practitioners; seeing where children are present and absent in utopian visions; examining how children can be respected and actively involved in shaping cities; and critically examining the "habitus in relation to children" [70] (p. 77). As discussed by Holloway and Valentine [66], the subfield of children's geography considers these social aspects of childhood and its influence on the construction and use of space, as well as the different sites of children's places within everyday life, the street, the playground, school, and home.

The inclusion of children in these processes relies on the conceptualization of childhood, as well as the other structures, norms, context, and the circumstance of the caretakers' or children's ability to participate. For example, younger age groups may require the participation of a caretaker. On the other hand, older children may be able to participate in more dialogue and have more independence [17]. Older children or adolescents may also be conceptualized as citizens, rather than just consultants to educate adults, which may result in more expressive and conversational participation approaches [17].

Further, Wood [58] draws from concepts of heterotopia and governmentality to explore how children provide a unique imaginary of space's potential, yet these imaginaries are often restricted by dominant ideas about children and planning. Geographic or spatial imaginaries, meaning the ways in which spaces are interpreted and perceived, or produce visions and discourse, influence courses of action [26,80] (p. 24). Further, Pérez del Pulgar et al. [75] considers influences on the "socio-natures" of children, seeking to uncover the processes of unseen interrelations between social and environmental systems in the production and experience of space, forming an understanding of children's "relational wellbeing."

Ataol, et al. [17] points to a set of methodologies often utilized in the practice of planning with children. These include diagnostic (analytical tools), expressive (creative

solutions), situational (collective situations), conceptual (abstract thinking), organizational, and political (e.g., child councils or forums) methods. Further expanding on organization and political methods, which aim to institutionalize children's inclusion, Chawla [61] also suggests training and investment in the facilitation of children's participation, providing budgets and community-based curricula, and adult commitment and facilitation [61] (p. 234–236). Burke [64] also highlights a "mosaic approach," which utilizes a mix of multiple methods to provide a more complex understanding of children's perspectives of their environments. These methods may include photographs, maps, child-led tours, dialogue, and observation [64].

Participatory approaches that allow children and youth to inform the planning process but also to be "change agents" provide several benefits, such as increased social awareness and skill development [60]. For example, methods that include children's perspectives and knowledge, acknowledge them as participants, and allow them to develop skills to influence change [81]. These processes can also transform the relationship between the adults and children, making them "co-creator[s] of knowledge" [60]. Further, using photovoice to supplement "formal" community health assessments, children felt empowered through the process of identifying their lived experiences [60]. Through participatory methods, children can also experience a sense of community and belonging, as found by Lam et al. [69] in using participatory approaches in school gardening. These connections allow children and youth to see themselves as part of the solution [67], while also providing a learning opportunity for adults about places for play [64].

Tracking and providing interventions across the life-course is also important as it relates to nature connection, and Capaldi et al. [12] suggests longitudinal studies to test the biophilia hypothesis and its influence. Also, combining research disciplines, including environmental psychology, urban geography, sociology, and children's geography, can assist in further understanding how children use green spaces [6]. Mixed-method approaches would also assist in considering multiple aspects related to the production of CES benefits [10]. For instance, Chawla [13] suggests applying both ethnographic and experimental/correlational designs to reveal how caretakers and children influence each other's experiences.

This review mostly included studies and literature originating from the United States and Europe, thus limiting the scope, and understanding of children's geographies, health inequities, and green space environmental justice. Future research could expand on these topics to include literature and studies from additional regions, particularly those in the Global South, where most of the world's children reside [82]. For instance, a review by Rigolon et al. [83] found that studies in the Global South show some similarities in trends and inequities related to green space to those on cities in the Global North. Additional research considering historical and current socio-economic and environmental factors influencing local development patterns and inequities is greatly needed.

**Author Contributions:** Conceptualization and writing—original draft preparation, M.V.; writing—review, editing and supervision, C.C. All authors have read and agreed to the published version of the manuscript.

**Funding:** This research received no external funding.

**Institutional Review Board Statement:** Not applicable.

**Informed Consent Statement:** Not applicable.

**Data Availability Statement:** Not applicable.

**Conflicts of Interest:** The authors declare no conflict of interest.

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
