# Peer review of "Considerations for Children’s Nature Connection and Park Environmental Justice in Western Societies"

_land, doi:10.3390/land11091435_

Round 1

Reviewer 1 Report

This review examines the elements of environmental justice in relation to children's playtime and connections to nature in nearby city parks. 

The paper is well written but the introduction should include relevant literature on  children's connection to nature during the COVID-19 pandemic, see for example https://besjournals.onlinelibrary.wiley.com/doi/full/10.1002/pan3.10270

https://www.nature.com/articles/d41586-022-00027-4

https://theconversation.com/1-in-2-primary-aged-kids-have-strong-connections-to-nature-but-this-drops-off-in-teenage-years-heres-how-to-reverse-the-trend-165660

https://www.tandfonline.com/doi/full/10.1080/03004279.2022.2052235

https://ijbnpa.biomedcentral.com/articles/10.1186/s12966-020-00987-8

https://pubmed.ncbi.nlm.nih.gov/35250152/

https://www.sciencedirect.com/science/article/pii/S004896972202188X

Reviewer 2 Report

Review of

Children’s Nature Connection & Park Environmental Justice: A 2 Review of the Literature

This is a potentially interesting paper with considerable value to the literature.

Methodology

I have a  major concern with  the methodology. There is now ample literature on how to construct and carry out systematic literature reviews. This paper does not provide a clear protocol for the choice of databases and keywords, nor any protocol for the processing (exclusion , retention) on the sources encountered. A statements (line 86)  like “Terms were varied and used in combination to refine searches of articles published between 2017-2021” is wholly inadequate. From line 91 it appears that the database searches were augmented by snowballing. Was that opportunistic or systematic. It would appear that the searches were confined to English-language sources but this is not spelled out. Did the search only cover refereed items, or also books? What about grey literature?. Was every item identified also evaluated, or were only those evaluated that could be readily accessed via the FSU library system and as free downloads via Google Scholar?

In short as described  in the paper the methodology chosen by the authors appears fundamentally flawed and cannot be replicated. Given this flaw, any further review of the paper is futile.

The authors are strongly encouraged to  place their methodology on a firm footing grounded in current standards for the conduct and reporting of systematic reviews. Based on this a re-evaluation of their collected sources may need to occur after which the authors are encouraged to resubmit

Reviewer 3 Report

This paper is of interest because it analyzes the components of environmental justice as they relate to children’s nature connection and play in local city parks. 

Nevertheless, it is too short and it lacks a wider reflection on the three following aspects: 

- the theme of making healthy places, https://islandpress.org/books/making-healthy-places-second-edition

- green cities and social injustice, https://www.routledge.com/The-Green-City-and-Social-Injustice-21-Tales-from-North-America-and-Europe/Anguelovski-Connolly/p/book/9781032024110 and http://www.bcnuej.org/wp-content/uploads/2021/04/Toolkit-Urban-Green-Justice.pdf 

-brownfields and green infrastructure projects, https://www.elgaronline.com/view/book/9781800375611/book-part-9781800375611-18.xml

These aspects should be thoroughly put in relation with children’s access to and activity in green spaces.

This is why I require a new version of the paper before its acceptance.

Reviewer 4 Report

The paper's subject is timely and concerns a Western society's problem: children spend less time outdoors than previous generations. A literature review on this subject is very useful. However, in my opinion, this paper is unsuitable for Land Journal. I would like to see this paper published in Education Sciences or IJERPH or Urban Science Journals.  

Additionally, the paper should be improved on minor topics.

Title: Please replace “Children’s Nature Connection & Park Environmental Justice: A Review of the Literature” by “Children’s Nature Connection & Park Environmental Justice: A Review of the Literature from Western societies”.

Lines 99-102: Please review this sentence. Even though “…the historic and persisting processes of racial segregation and oppressions…” are true in the US, it isn't necessarily true for other western countries.

Line 212: Please replace “…Rigolon & Flohr [26]…” by “…Rigolon & Flohr [27]…”

Lines 235-237: Please review this sentence. Although the one-child notice is important, this is not sufficiently representative and should be better contextualized.

Line 275: Please replace “…Gillespi [58]…” by “…Gillespie [58]…”

Line 280: Please replace “…Gillespi [58]…” by “…Gillespie [58]…”

Lines 333-334: Please replace “…Gillespie (2012) suggests…” by “…Gillespie (2013) suggests…”

Line 339: Please replace “…Holloway (2004), the…” by “…Holloway & Valentine (2004), the…”

Line 390: Please explain the CES meaning

Line 515: Please replace “49. Louise Chawla Growing up…” by “49. Chawla, L. Growing up...”

Round 2

Reviewer 2 Report

I am satisfied with the changes made

Author Response

We are grateful for your time and detailed feedback, which has significantly strengthened this paper.

Reviewer 3 Report

The paper has been revised but not all the works suggested to cite have been cited, so please do it for the international readership.

Author Response

Point 1: The paper has been revised but not all the works suggested to cite have been cited, so please do it for the international readership.

Response 1: Thank you for providing further clarification on the suggested revision. We agree that the addition of the specific citations you provided further assist in situating these topics within a more international setting. Revisions were made in the introduction on lines 22-47 to include reference to these citations.